# SpikingVTG: A Spiking Detection Transformer for Video Temporal Grounding

**Malyaban Bal**[1,2][*], **Brian Matejek**[2], **Susmit Jha**[2], **Adam D. Cobb**[2]

[1]The Pennsylvania State University
[2]Computer Science Laboratory, SRI International

## Abstract

Video Temporal Grounding (VTG) aims to retrieve precise temporal segments in a video conditioned on natural language queries. Unlike conventional neural frameworks that rely heavily on computationally expensive dense matrix multiplications, Spiking Neural Networks (SNNs)—previously underexplored in this domain—offer a unique opportunity to tackle VTG tasks through bio-plausible spike-based communication and an event-driven accumulation-based computational paradigm. We introduce SpikingVTG, a multi-modal spiking detection transformer, designed to harness the computational simplicity and sparsity of SNNs for VTG tasks. Leveraging the temporal dynamics of SNNs, our model introduces a Saliency Feedback Gating (SFG) mechanism that assigns dynamic saliency scores to video clips and applies multiplicative gating to highlight relevant clips while suppressing less informative ones. SFG enhances performance and reduces computational overhead by minimizing neural activity. We analyze the layer-wise convergence dynamics of SFG-enabled model and apply implicit differentiation at equilibrium to enable efficient, BPTT-free training. To improve generalization and maximize performance, we enable knowledge transfer by optimizing a Cos-L2 representation matching loss that aligns the layer-wise representation and attention maps of a non-spiking teacher with those of our student SpikingVTG. Additionally, we present Normalization-Free (NF)-SpikingVTG, which eliminates non-local operations like softmax and layer normalization, and an extremely quantized 1-bit (NF)-SpikingVTG variant for potential deployment on edge devices. Our models achieve competitive results on QVHighlights, Charades-STA, TACoS, and YouTube Highlights, establishing a strong baseline for multi-modal spiking VTG solutions.

## 1 Introduction

The rapid expansion of various social medias and portable smart technologies has triggered an unprecedented surge in video content. This vast influx of data has intensified the need for efficient methods to retrieve and analyze video information. Consequently, the field of Video Temporal Grounding (VTG) [1] has emerged as an important area of research. The main objective of VTG is to identify the precise segment of a video that corresponds to a given natural language query, enabling accurate and context-driven video content retrieval. In this paper, we focus on two tasks: moment retrieval [2, 3], which aims to identify video intervals relevant to a given query, and highlight detection [4], which retrieves the best candidate segment of the video in response to the query. Our work involves analyzing multimodal data—combining video content with natural language queries—to develop an effective solution to the problem. With the rise of transformer based architectures the field of VTG has seen significant advancements [5, 6]. However, these models demand substantial

---

[*]Work completed while at SRI.

39th Conference on Neural Information Processing Systems (NeurIPS 2025).

power and energy [7] to operate. Furthermore, VTG is inherently resource-intensive, requiring the analysis of long video sequences, leading to significant computational overhead. Additionally, applications such as **edge-based event detection** in video—e.g., identifying accidents from traffic cameras—require deploying VTG models on resource-constrained devices near the data source, in order to reduce data transfer to the cloud. These devices often operate under limited energy availability. Inspired by neural dynamics in the brain, this paper leverages bio-plausible neuronal models and learning dynamics to develop an efficient and brain-inspired solution for VTG tasks. Our model enables flexible **inference-time tradeoff** between **energy consumption and accuracy**, making it well-suited for **edge-based deployment in VTG tasks**.

**SpikingVTG Model:** We introduce SpikingVTG, a spiking detection transformer designed for efficient and accurate VTG. Built on the sparse, event-driven communication and accumulation-based computation of Spiking Neural Networks (SNNs) [8], SpikingVTG leverages the intrinsic dynamics of SNNs to offer a lightweight yet competitive alternative to conventional transformer-based approaches—eliminating the need for costly dense, real-valued matrix multiplications. The architecture comprises three key components: (i) a spiking transformer core, (ii) a Saliency Feedback Gating (SFG) mechanism, and (iii) a spiking decoder for output generation. The spiking transformer captures temporal and cross-modal dependencies, while the decoder produces task-specific outputs. The SFG mechanism addresses a critical challenge unique to VTG: given an input video composed of many clips, how can we identify those most relevant to the query? Tailored specifically for multi-modal VTG tasks, the SFG module leverages the temporal dynamics of SNNs to attend more towards the most salient segments while suppressing irrelevant clips.

**Saliency Feedback Gating Mechanism:** Operating over discrete time steps, SpikingVTG uses the intermediate spiking activity of the transformer core to dynamically estimate the relevance of each video segment. We compute a feedback-based saliency score for each segment based on the *average spiking rate* (ASR) of the output of the transformer, conditioned on the query. These scores serve as soft attention masks within a multiplicative gating mechanism, suppressing less informative segments to reduce computational overhead while enhancing focus on the most relevant candidate clips. From a neuroscience perspective, **feedback connections** are known to play a critical role in object recognition in the visual cortex [9]. Furthermore, our **feedback mechanism maintains layer-wise convergence of ASR** to equilibrium, enabling us to adopt an efficient training mechanism leveraging the equilibrium dynamics [10]. This approach circumvents the need for computationally intensive BPTT [11], and instead updates model parameters using a single backward pass, significantly improving memory efficiency.

**Cos-L2 Representation Matching (CLRM):** While our base SpikingVTG model achieves performance comparable to non-pretrained, non-spiking VTG models such as M-DETR and UniVTG [1, 6], achieving state-of-the-art results on VTG tasks typically requires transformer-based, non-spiking models like UniVTG to undergo extensive pre-training. This pre-training significantly boosts their generalization and task-specific capabilities. To enable our SpikingVTG to benefit from similar pre-training advantages—without incurring the high computational cost of training on large-scale datasets like Ego4D, Video-CC [12]— we propose a direction and scale aware CLRM loss for efficient knowledge transfer. Optimizing CLRM loss aligns the hidden states and attention score maps of a pre-trained non-spiking multi-modal transformer with the hidden state converged ASR and mean attention scores of our SpikingVTG model. By minimizing this alignment loss on the downstream task, we allow the student model to imbibe the generalization capability learnt by the pre-trained teacher without having to perform the extensive pre-training from scratch.

**Optimizations and Application to VTG Tasks:** Traditional transformer-based VTG solutions [6, 13] rely heavily on non-local normalization operations, such as softmax and layer normalization, which pose significant challenges for efficient implementation on neuromorphic hardware [14]. To address this limitation, we develop and evaluate a Normalization-Free (NF)-SpikingVTG model, which eliminates all layer normalization operations and substitutes softmax spiking attention with a ReLU and scaling-based spiking attention mechanism. Although alternative Softmax-free attention approaches have been explored in the literature [15, 16], they have primarily been applied uni-modal vision tasks. We empirically demonstrate that while these optimizations enhance computational efficiency, they result in minimal performance degradation.

To further **reduce computational complexity** and **memory footprint** [17, 18], we introduce 1-bit (NF)-SpikingVTG model, which rely primarily on **integer accumulation operations** and **eliminate**

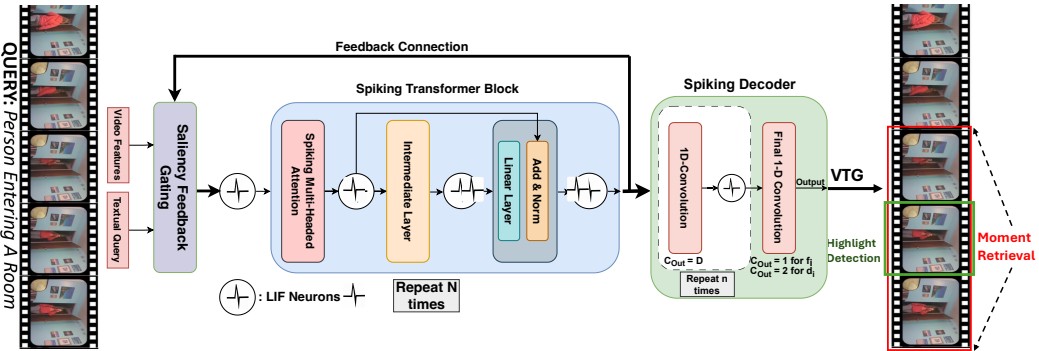

Figure 1: High-level overview of the proposed SpikingVTG model. The model employs a spiking transformer core that utilizes Saliency Feedback Gating through temporal feedback connections. The model also incorporates a spiking decoder module that takes the output of the transformer core to predict parameters for the VTG task.

**all non-local operations**. This design makes it well-suited for deployment on resource-constrained edge devices. To our knowledge, this work is the first to evaluate an operational spiking detection transformer across VTG tasks, including moment retrieval and highlight detection, on datasets such as QVHighlights, Charades-STA, TACoS and Youtube Highlights. To further highlight the benefits of using SpikingVTG, we present an energy-accuracy tradeoff analysis. This demonstrates that our model is well-suited for deployment on edge devices, where its performance can be dynamically adjusted based on the available energy budget.

## 2 Related Works

**VTG Advancements:** Moment-DETR [1], a transformer encoder-decoder model introduced alongside the QVHighlights dataset, laid a strong foundation for subsequent VTG architectures. UMT [5] introduced an unified framework for solving both highlight detection and moment retrieval tasks. Due to the limited availability of trainable video data, UniVTG [6] proposed an innovative solution by unifying various VTG tasks and labels under a single formulation. This enables the development of an LLM-like pretraining framework, achieving state-of-the-art performance on VTG tasks. Although no fully spiking-based model has been explored for VTG tasks, SpikeMba [19]—primarily a non-spiking model—integrates SNN components to generate proposal sets from video data. However, since its core framework is derived from Mamba [20] and relies on floating-point matrix multiplications, SpikeMba cannot be considered a baseline for spiking models, which predominantly use accumulation-based operations. While recent VTG models have significantly improved task-specific performance, adopting a spiking framework enables us to leverage energy-accuracy tradeoff—making suitable solutions on edge devices with limited or dynamic energy supply.

**Spiking neural networks (SNNs):** SNNs have been implemented in neuromorphic systems like IBM TrueNorth [21] and Intel Loihi 2 [22], demonstrating approximately $75\times$ greater energy efficiency compared to traditional networks running on low-power GPUs [23]. SNNs, with their energy-efficient computational framework, offer a promising solution to the resource-intensive demands of multimodal VTG tasks. While SNNs for a long time were confined in simpler vision-based tasks [24] with simple architectures, recent developments have scaled them to transformer-based models for tasks ranging from vision to language modelling [25, 26, 27], however majority of them are uni-modal and rely on non-local operations not implementable on a neuromorphic chip.

## 3 Video Temporal Grounding (VTG)

For a given video $V$ and language query $Q$, we start by segmenting $V$ into a sequence of $L_v$ fixed-length clips, denoted as $\{v_1, \ldots, v_{L_v}\}$. Each clip $v_i$ has a length $l$ and is centered at timestamp $t_i$. The textual query $Q$ consists of $L_q$ tokens, denoted as $Q = \{q_1, \ldots, q_{L_q}\}$. Following previous studies on VTG [6], we define three parameters for each clip $v_i = (f_i, d_i, s_i)$, where $f_i = 1$ if the

clip is in foreground, i.e. relevant else $f_i = 0$. $d_i = [d_{s_i}, d_{e_i}] \in \mathbb{R}^2$ represent the temporal distance that converts the clip timestamp $t_i$ to its interval boundaries. Here, $d_i$ is valid when $f_i = 1$. The term $d_{s_i}$ denotes the distance between the start of the interval and $t_i$, while $d_{e_i}$ denotes the distance between the end of the interval and $t_i$. $s_i \in [0, 1]$ is a continuous score that quantifies the relevance between the visual content of clip $v_i$ and the query $Q$. Our proposed SpikingVTG predicts these three parameters for each video clip. In this paper, we focus on the following VTG tasks:

**Moment Retrieval:** We rank the predicted clip boundaries $\{\tilde{b}_i\}_{i=1}^{L_v}$, where $b_i = [t_i - d_{s_i}, t_i + d_{e_i}]$, based on their associated probabilities given by $\{\tilde{f}_i\}_{i=1}^{L_v}$. Since the predicted $L_v$ boundaries are dense, we employ a 1-dimensional Non-Maximum Suppression (NMS) [28] with a threshold of 0.7 to eliminate highly overlapping boundary boxes, resulting in a final prediction.

**Highlight Detection** For each clip, we rank all clips based on their combined scores $\{\tilde{f}_i + \tilde{s}_i\}_{i=1}^{L_v}$. This combined value represents how well the chip $i$ match with the underlying query. We then return the top clips (e.g., Top-1) as predictions.

# 4 SpikingVTG: Architecture Overview

The core computational unit of the proposed SpikingVTG model is a leaky integrate-and-fire (LIF) neuron [29]. Neurons communicate with each other using sparse, spike-based activations instead of real-valued signals, thus we can replace floating-point matrix multiplications with accumulative operations, resulting in improved computational efficiency.

## 4.1 Spiking Neural Networks

The discrete time dynamics of an LIF-based spiking neuron can be given as follows,

$$
\begin{aligned}
u_i[t + \delta] &= \gamma u_i[t] + W_{(i-1)}(s_{(i-1)}[t]) + b_i, \\
u_i[t + 1] &= u_i[t + \delta] - V_{th_i} s_i[t + 1], \\
a_i[t] &= \frac{\sum_{\tau=1}^{t} \gamma^{t-\tau} s_i[\tau]}{\sum_{\tau=1}^{t} \gamma^{t-\tau}}.
\end{aligned}
\tag{1}
$$

where, at time $t$, $u_i[t]$ is the membrane potential of the $i^{th}$ neuronal layer; $b_i$ indicates a bias term and $\gamma$ is the leaky term. $W_{(i-1)}$ represents the layer-specific operation; $t + \delta$ is an intermediate time step to determine if the neuron fired; $V_{th_i}$ is the threshold of layer $i$. We use a ternary spiking model [30] in our work for spike ($s[t + 1]$) generation. This improves performance while avoiding the introduction of additional floating-point multiplicative and accumulative (FP-MAC) operations. The average spiking rate (ASR $a_i[t]$) of neurons within each layer $i$ at time $t$ can be defined as a weighted-average function.

## 4.2 Spiking Transformer Core

The high-level overview of each encoder block of our spiking transformer architecture is demonstrated in Fig. 1. The model consists of $N$ encoder layers, each consists of a spiking multi-headed attention block, followed by an intermediate layer and an output layer. Communication within and between encoder layers occurs via spikes. Furthermore, all matrix multiplications involved in linear layers and attention layer comprises of more efficient fp-accumulative (FP-ACC) operations instead of FP-MAC operations in conventional neural architectures. Detailed descriptions of each layer are provided in the Appendix A. Following architectural optimizations (Section 4.6), we replace softmax-based attention with a ReLU and scaling-based spiking attention mechanism, remove all layer normalization operations, and explore extreme quantization of linear weights.

## 4.3 Saliency Feedback Gating (SFG)

SpikingVTG operates over a specific number of convergence time steps ($T_c$), with the convergence dynamics detailed in Section 4.5. This temporal processing allows us to leverage intermediate temporal outputs to dynamically update the input to the model at every time step for improved performance. This approach conforms to the feedback connections observed in the human visual

cortex [9], providing a bio-plausible explanation for its efficacy. The ASR of the final encoder layer of the Spiking Transformer core is used as a temporal feedback to compute a dynamic saliency score with the input query enabling the design of a gating mechanism (Fig. 2a), allowing selective focusing on relevant segments of the video while minimizing computation on irrelevant segments. The saliency feedback gating mechanism is shown below,

$$
\begin{aligned}
F_s^{v_i}[t] = \cos(\mathbf{a_{N_v}^i}[\mathbf{t}], \mathbf{M}) &:= \frac{\mathbf{a_{N_v}^i}[\mathbf{t}] \cdot \mathbf{M}}{\|\mathbf{a_{N_v}^i}[\mathbf{t}]\|_2 \|\mathbf{M}\|_2}, \\
\bar{V}[t+1] &= V[t] * \bar{F}_s^v[t], \\
\mathbf{SFG}(V, \mathbf{Q}, \mathbf{a_{N_v}}) &= \bar{V}[t+1] \oplus \mathbf{Q},
\end{aligned}
\tag{2}
$$

where, using attentive pooling operation, sentence representation $\mathbf{M} = \mathbf{Q}^T Softmax(\mathbf{QW_P})$, $\mathbf{M} \in \mathbb{R}^D$, input textual query features $\mathbf{Q} \in \mathbb{R}^{L_q \times D}$, input video features $V \in \mathbb{R}^{L_v \times D}$ and $\mathbf{W_P} \in \mathbb{R}^{D \times 1}$ is a learnable embedding and $D$ is the hidden dimension. $F_s^{v_i}[t]$ is the dynamic saliency score, at time $t$, for the $i$-th segment of the video. We compute $\bar{F}_s^v$ by applying min-max normalization to $F_s^v$, allowing us to obtain per-clip scores within the range $[0, 1]$. The ASR of the output of the spiking transformer core is given as $a_N[t] \in \mathbb{R}^{(L_v + L_q) \times D}$. $a_{N_v}^i[t]$ is ASR of output of the spiking transformer core, corresponding to video segment $i$, at time $t$. The output of SFG module is the concatenation of saliency feedback gated video features and query features and serves as the input to the spiking transformer core at time $t+1$.

To ensure compatibility between the two modalities during cosine similarity computation, both representations are projected into a shared latent space of dimensionality $D$. The query representation is aggregated into a global sentence embedding via attention-based pooling with a learnable embedding vector. This adaptive mapping aligns the textual representation with the spatiotemporal semantics of the spiking video features, thereby facilitating effective cross-modal compatibility.

**SFG Computational Overhead:** The SFG layer comprises $O(L_v \cdot D)$ floating-point multiplication operations; however, the computational overhead of this layer is significantly less than that of the transformer core which has a complexity of $O(L^2 \cdot D + L \cdot D^2)$, where $L = L_v + L_q$.

### 4.3.1 Visualizing effect of SFG:

The dynamic saliency score ($F_s^v$) achieves an equilibrium value $F_s^*$ following the convergence of ASR of the neuronal layers of the SpikingVTG model (Fig. 3). As shown in Fig. 2b, we empirically analyze the scores per clip at equilibrium to gain insights into the functioning of the SFG based multiplicative gating mechanism. Neighboring salient clips of the target clip exhibit higher $\bar{F}_s$ scores at equilibrium, while irrelevant clips show lower scores, highlighting the effectiveness of the SFG mechanism.

The SFG mechanism not only results in **better performance** of our SpikingVTG architecture (Table 5) but also **reduces overall neural activity** by sparsifying input spikes. Empirical

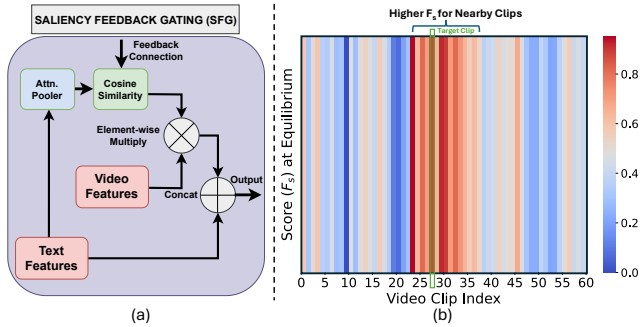

Figure 2: (a) Overview of the internal operations of the saliency-feedback gating mechanism. The ASR of the output of the spiking transformer core at each time step is leveraged as the feedback signal (Fig. 1). (b) Heatmap showing the scores per clip ($\bar{F}_s$) at equilibrium, with the target frame for highlight detection corresponding to clip index 28.

results (Fig. 3b) confirm that the model with the gating mechanism exhibits a lower neural activity, particularly in the input and spiking attention layers, compared to the model without SFG.

### 4.4 Spiking Decoder Module

The spiking decoder comprises of stacked 1-D convolutions followed by integrate-fire (IF) neuron layers ($\gamma = 1$ in Eqn. 1), for spike generation. The spiking decoder used for predicting foreground indicator ($f_i$) per clip, applies $n_1$ 1-D convolution operations with kernel size $k_1$, each followed by an

IF layer. The final layer consists of a single output channel, and its temporal mean is passed through a sigmoid activation to produce the prediction. The spiking decoder used for $d_i$ applies $n_2$ 1-D convolution operations with kernel size $k_2$, each followed by an IF layer, and the final convolution layer has two output channels to predict $d_i = [d_{s_i}, d_{e_i}]$, after which we compute $b_i$.

## 4.5 SpikingVTG: Convergence Dynamics

The membrane potential ($u_1$) of the input LIF layer to the spiking transformer core, following the SFG mechanism can be formulated as below,

$$u_1[t+1] = \gamma u_1[t] + \mathbf{SFG}(V, Q, a_{N_v}[t]) + b_1 - V_{th_1} s_1[t+1] \tag{3}$$

where, $a^{N_v}[t]$ is the ASR of the final layer corresponding to the video features at time $t$, $V$ is the input video features, $Q$ is the query features and $\mathbf{SFG}$ is defined in Eqn. 2. The layer-wise convergence dynamics of the SpikingVTG with SFG is demonstrated in Fig. 3.

Following Eqn. 1, the layer-wise ASR is, $a_i[t+1] = \frac{1}{V_{th_i}}(\hat{f}(a_{(i-1)}[t+1]) + b_i - \frac{u_i[t+1]}{\sum_{j=0}^{t} \gamma^j})$ where, $\hat{f}$ is operation of layer $i$. Following empirical evidence (Fig. 3) and theoretical formulation [10, 26] as time $t \to \infty$, the layer-wise ASRs converge to equilibrium, enabling the derivation of layer-wise steady-state equations given as,

$$a_i^* = \sigma(\frac{1}{V_{th_i}}(\hat{f}(a_{i-1}^*) + b_i)) \tag{4}$$

The equation describing the steady-state ASR dynamics of the input layer is given as,

$$a_1^* = \sigma(\frac{1}{V_{th_1}}(\mathbf{SFG}(V, Q, a_{N_v}^*) + b_1)) \tag{5}$$

where, clipping function $\sigma(x)$ clamps the values within $[-1, 1]$. This is because we allow ternary spikes thus ASR must be with $[-1, 1]$. Furthermore, the dynamic saliency score also achieves an equilibrium value $F_s^*$ since the ASR at the final layer achieves equilirbrium state $a_{N_v}^*$. Convergence dynamics of other layers are given in Appendix A. To analyze the overall layer-wise neural activity, which includes both positive and negative spiking event, we present the layer-wise dynamics of the absolute spiking activity events in Fig. 3b, i.e. $act_i[t] = \frac{\sum_{i=1}^{t} |s_i[t]|}{t}$.

**Training:** As described in the Section 4.4, the Spiking decoder is responsible for predicting $\tilde{f}_i$ and $\tilde{d}_i$ for individual video clip $i$ and $\tilde{s}_i$ is computed using the SFG module at equilibrium. Using these three predictions, we design a loss function that combines various components. The total loss over $N_T$ clips in the training set is given by $L = \frac{1}{N_T} \sum_{i=1}^{N_T} (L_{f_i} + L_{d_i} + L_{c_i})$, where $L_f$ is the binary cross-entropy loss for the indicator variable $f_i$, $L_d$ combines smooth L1 loss with IoU loss [31] for the predicted boundaries, and $L_c$ is an optional loss incorporating intra- and inter-video contrastive learning [32]. Detailed formulation of the loss functions is in Appendix B.

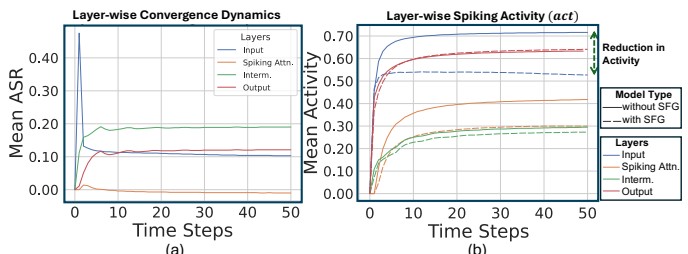

Figure 3: Results from a random QVHighlights input passed through SpikingVTG models. (a) Layer-wise mean ASR convergence over operating time steps for a sample spiking transformer encoder layer (Fig. 1); note ASR can be negative due to ternary spikes. (b) Layer-wise mean spiking activity ($act_i[t]$, averaged over neurons) versus time steps. Model with SFG exhibit reduced activity in both input and spiking attention layers, highlighting SFG's role in minimizing neural activity.

During training, leveraging implicit differentiation [33] at equilibrium, only ASR values at equilibrium are used, $\frac{\partial L(a^*)}{\partial \theta} = -\frac{\partial L(a^*)}{\partial a^*}(J_{g_\theta}^{-1}|_{a^*})\frac{\partial f_\theta(a^*)}{\partial \theta}$ where, $\theta$ is the model parameters, $g_\theta(a) = f_\theta(a) - a$,

$f$ is the steady-state equation of ASR, $J^{-1}$ is the inverse Jacobian of $g_\theta$ when $a = a^*$, i.e., at equilibrium. Thus, unlike BPTT, we do not need to store the intermediate computational graph and the model parameters can be updated using a single backpropagation step.

#### 4.5.1 Cos-L2 Representation Matching (CLRM)

We train the base SpikingVTG model on VTG tasks such as QVHighlights [1], achieving comparable performance to a similarly scaled non-spiking transformer-based model, UniVTG [6], as demonstrated in Table 1. The later to improve its performance further uses pre-training on large datasets such as Video-CC and Ego4D. Since, extensive pretraining of the SpikingVTG is considerably resource intensive we propose an effective knowledge transfer strategy. This mechanism enables SpikingVTG to inherit the generalization capabilities learned by the pre-trained detection transformer models like UniVTG model. Importantly, since the primary benefit of spiking architectures manifests during inference on resource-constrained edge devices, this knowledge transfer is a one-time process, which significantly improves performance.

**Hidden State Matching Loss:** To align the hidden state (output of individual encoder layer) of the non-spiking multi-modal transformer model with the converged ASR (4.5) of the corresponding layer of the SpikingVTG model, we propose a hybrid loss function combining a

Table 1: **Ablation study of the effect of CLRM** on SpikingVTG evaluated on the evaluation set of QVHighlights.

| Method | QVHighlights-MR | | | QVHighlights-HD | |
|---|---|---|---|---|---|
| | @0.5 | @0.7 | mAP@avg | mAP | HIT@1 |
| UniVTG [6] | 59.74 | 40.90 | 36.13 | 38.83 | 61.81 |
| SpikingVTG w/o CLRM | 60.12 | 39.68 | 36.23 | 38.84 | 62.49 |
| UniVTG with PT [6] | 67.35 | 52.65 | 45.44 | 41.34 | 68.77 |
| SpikingVTG w/ CLRM | **67.58** | **50.82** | **44.07** | **40.81** | **68.64** |

squared cosine similarity based directional loss and a L2-norm based loss for minimizing the scale difference. For each layer, the student representation is projected into the feature space of the teacher via a learnable linear transformation $W_d \in \mathbb{R}^{d_s \times d_t}$. The total loss is:

$$\mathcal{L}_{\text{rep}} = \frac{1}{B \times L} \sum_{i=1}^{N} \sum_{j=1}^{B} \sum_{k=1}^{L} \left[ \lambda_{\cos} \cdot \left( 1 - \cos\left( \theta_{i,j,k}^{\text{rep}} \right) \right)^2 + \lambda_{\ell_2} \cdot \left\| \mathbf{s}_i^{(j,k)} - \mathbf{t}_i^{(j,k)} \right\|_2^2 \right] \qquad (6)$$

where $N$ is total number of encoder layers, $B$ is batch size, $L$ is length of sequence, $\lambda_{\cos}, \lambda_{l_2}$ are hyperparameters and the cosine similarity is computed as: $\cos(\theta_{i,j,k}^{\text{rep}}) = \frac{\mathbf{s}_i^{(j,k)} \cdot \mathbf{t}_i^{(j,k)}}{\|\mathbf{s}_i^{(j,k)}\|_2 \cdot \|\mathbf{t}_i^{(j,k)}\|_2}$, where, $\mathbf{s}_i^{(j,k)} = a_{r_i}^{*(j,k)} W_d \in \mathbb{R}^{d_t}$ is the projected student representation and $\mathbf{t}_i^{(j,k)} = T_{r_i}^{(j,k)} \in \mathbb{R}^{d_t}$ is the teacher representation at layer $i$, sequence position $k$, and batch index $j$. $a_{r_i}^* \in \mathbb{R}^{B \times L \times d_s}$ and $T_{r_i} \in \mathbb{R}^{B \times L \times d_t}$ are the pre-activation outputs from student and teacher, respectively.

**Attention Score Matching Loss:** We align the attention score map ($\mathbf{A_i^T} \in \mathbb{R}^{B \times L \times L}$) of encoder layer $i$ of the non-spiking model with the mean attention score ($\mathbf{A_i^{S^*}} = \frac{1}{T_c} \sum_{t=1}^{T_c} A_i^S[t] \in \mathbb{R}^{B \times L \times L}$) of our corresponding converged SpikingVTG model. The total attention map alignment loss ($\mathcal{L}_{\text{att}}$) is then computed following Eqn. 6. This loss term encourages the student to learn cross-modal attention behavior consistent with the pre-trained teacher.

The cumulative representation matching loss is given as $\mathcal{L}_{\text{CLRM}} = \mathcal{L}_{\text{rep}} + \mathcal{L}_{\text{att}}$. Optimizing this loss enables our SpikingVTG model to learn the generalability enabling better performance as demonstrated in Table 1.

### 4.6 Normalization-Free and Quantized SpikingVTG

We perform two key optimizations: removal of non-local operations and 1-bit weight quantization. Although SpikingVTG replaces floating-point MACs with ternary spikes, it retains softmax and layer normalization, which are inefficient for resource-constrained hardware. We eliminate these by introducing a Normalization-Free (NF) SpikingVTG.

In NF-SpikingVTG, softmax in the spiking attention is replaced with a ReLU followed by scaling with $1/L$, reducing compute overhead. Given $d$-dimensional queries, keys, and values $\{q_i[t], s_{k_i}[t], s_{v_i}[t]\}_{i=1}^{L}$, at time $t$, the attention weights $\alpha_{ij}$ are computed as follows:

$$\alpha_{ij}[t] = \phi \left( \left[ q_i[t]^\top s_{k_1}[t], \cdots, q_i[t]^\top s_{k_L}[t] \right] \right)_j \qquad (7)$$

where, $\phi$ is a custom kernel consisting of ReLU operation followed by scaling with $L^{-1}$. We further remove all layer normalization layers to eliminate additional non-local operations.

To reduce memory and computational cost, we introduce 1-bit NF-SpikingVTG by binarizing weights. Each weight matrix $W \in \mathbb{R}^{n \times m}$ is zero-centered and quantized as $W_q = \text{sgn}\left(W - \frac{1}{nm}\sum_{i,j} W_{ij}\right)$, $\beta = \frac{1}{nm}\sum_{i,j}|W_{ij}|$, where $W_q \in \{-1, +1\}$. The output of each quantized linear layer is scaled by $\beta$, yielding a binary-weight, ternary-activation model that supports efficient integer-ACC operations. 1-bit NF-SpikingVTG achieves competitive performance while drastically reducing memory and compute, making it well-suited for edge deployment.

Table 2: Performance comparison of our SpikingVTG model against non-spiking VTG solutions on **test sets** of QVHighlights-MR and Charades-STA datasets for moment retrieval task.

| Method | SNN | QVHighlights-MR | | | | Charades-STA | | | |
|---|---|---|---|---|---|---|---|---|---|
| | | @0.5 | @0.7 | mAP@0.5 | mAP@0.75 | @0.3 | @0.5 | @0.7 | mIoU |
| M-DETR [1] | No | 52.89 | 33.02 | 54.82 | 29.40 | 65.83 | 52.07 | 30.59 | 45.54 |
| UMT [5] | No | 56.23 | 41.18 | 53.83 | 37.01 | - | 49.35 | 26.16 | - |
| UniVTG [6] | No | 58.86 | 40.86 | 57.60 | 35.59 | 70.81 | 58.01 | 35.65 | 50.10 |
| UniVTG+PT [6] | No | 65.43 | 50.06 | 64.06 | 45.02 | 72.63 | 60.19 | 38.55 | 52.17 |
| UVCOM [34] | No | 63.55 | 47.47 | 63.37 | 42.67 | - | 56.69 | 34.76 | - |
| SpikeMba [19] | No | 64.13 | 49.42 | - | 43.67 | 71.24 | 59.65 | 36.12 | 51.74 |
| BAM-DETR [35] | No | 62.71 | 48.64 | 64.57 | 46.33 | 72.93 | 59.95 | 39.38 | 52.33 |
| TR-DETR [36] | No | 64.66 | 48.96 | 63.98 | 43.73 | - | 57.61 | 33.52 | - |
| CG-DETR [37] | No | 65.43 | 48.38 | 64.51 | 42.77 | 70.40 | 58.40 | 36.30 | 50.10 |
| LLMEPET [38] | No | 66.73 | 49.94 | 65.76 | 43.91 | 70.91 | - | 36.49 | 50.25 |
| **SpikingVTG** | **Yes** | **65.29** | **48.18** | **64.31** | **42.25** | **71.20** | **58.73** | **37.16** | **50.62** |

## 5 Experimentation

We evaluate SpikingVTG variants on moment retrieval and highlight detection tasks using the QVHighlights, Charades-STA, TACoS and Youtube Highlight datasets. Since, to the best of our knowledge, our proposed model is the first spiking detection transformer evaluated on VTG tasks, we compare its performance against sota non-spiking detection transformers. Additional training, dataset, evaluation metric, hyperparameter and experimental details are provided in Appendix C & E. The experiments were run on a NVIDIA RTX A6000 GPU with 48GB memory.

Table 3: Performance comparison on the **test set** of TACoS for **moment retrieval task**.

| Method | SNN | TACoS | | | |
|---|---|---|---|---|---|
| | | @0.3 | @0.5 | @0.7 | mIoU |
| M-DETR [1] | No | 37.97 | 24.67 | 11.97 | 25.49 |
| 2D-TAN [6] | No | 40.01 | 27.99 | 12.92 | 27.22 |
| UniVTG [6] | No | 51.44 | 34.97 | 17.35 | 33.60 |
| QD-DETR [39] | No | 52.39 | 36.77 | 21.07 | 35.76 |
| CG-DETR [37] | No | 52.23 | - | 22.23 | 36.48 |
| UniVTG+PT [6] | No | 56.11 | 43.44 | 24.27 | 38.63 |
| SpikeMba [19] | No | 51.98 | 39.34 | 22.83 | 35.81 |
| **SpikingVTG** | **Yes** | **54.71** | **39.27** | **21.84** | **36.02** |

### 5.1 Results

SpikingVTG establishes a baseline for spiking models on VTG tasks. The results are shown in Table 2, 3 & 4. Our model achieves competitive results compared to the current SOTA non-spiking models.

**Training & Inference Metrics:** In the CLRM-based knowledge transfer stage, the memory requirement is 20GB when using a batch size of 32 on the QVHighlights dataset. In contrast, replacing our training method with BPTT would require over 100GB of memory for $T_c > 10$, making BPTT computationally infeasible. Training on true labels with similar batch size requires 8GB of

Table 4: Performance comparison of our SpikingVTG model against other non-spiking VTG solutions on the **test set** of the QVHighlights and Youtube Highlights for **highlight detection task**.

| Method | SNN | QVHighlights-HD | | Youtube Highlights | | | | | | |
|---|---|---|---|---|---|---|---|---|---|---|
| | | mAP | HIT@1 | Dog | Gym. | Skating | Skiing | Parking | Surfing | Avg. |
| UMT [5] | No | 38.18 | 59.99 | 65.9 | 75.2 | 71.8 | 72.3 | 81.6 | 82.7 | 74.9 |
| UniVTG [6] | No | 38.20 | 60.96 | 71.8 | 76.5 | 73.3 | 73.2 | 73.9 | 82.2 | 75.2 |
| UniVTG+PT [6] | No | 40.54 | 66.28 | 74.3 | 79.0 | 84.9 | 75.1 | 74.4 | 83.9 | 78.6 |
| QD-DETR [39] | No | 38.90 | 62.40 | 72.2 | 77.4 | 72.7 | 72.8 | 71.0 | 80.6 | 74.4 |
| SpikeMba [5] | No | - | - | 74.4 | 75.4 | 74.3 | 75.5 | - | - | 75.5 |
| CG-DETR [37] | No | 40.30 | 66.20 | 76.3 | 76.1 | 76.0 | 75.1 | 70.0 | 81.9 | 75.9 |
| UVCOM [34] | No | 39.98 | 65.58 | 73.8 | 77.1 | 76.0 | 75.1 | 75.7 | 82.7 | 76.4 |
| LLMEPET [38] | No | 38.18 | 59.99 | 73.6 | 73.2 | 75.3 | 74.0 | 72.5 | 82.5 | 75.3 |
| **SpikingVTG** | **Yes** | **40.46** | **65.82** | **73.9** | **78.1** | **80.1** | **74.2** | **72.2** | **81.7** | **76.7** |

memory. The clock time for 50 epochs of training on QVHighlights is $\approx 5$ HOURS. Inference latency on entire test set of QVHighlights ranges from 25 sec ($T_c = 2$) to $\approx 4$ mins ($T_c = 20$).

Table 5: **Ablation Study** of SpikingVTG variants on the evaluation set of QVHighlights.

| Method | QVHighlights-MR | | QVHighlights-HD | | Operations | Local | Activity | Energy |
|---|---|---|---|---|---|---|---|---|
| | @0.5 | @0.7 | mAP | HIT@1 | | | | |
| Pre-trained UniVTG (sota) | 67.35 | 52.65 | 41.34 | 68.77 | FP-MAC | × | 1.0 | 23.92mJ |
| SpikingVTG **without SFG** | 64.94 | 47.21 | 40.49 | 67.37 | FP-ACC | × | 0.41 | 15.2mJ |
| SpikingVTG **with SFG** | **67.58** | **50.82** | **40.81** | **68.64** | FP-ACC | × | 0.34 | 13.8mJ |
| (NF)-SpikingVTG w/ SFG | 66.59 | 48.31 | 40.61 | 67.73 | FP-ACC | ✓ | 0.25 | 10.1mJ |
| 1-bit (NF)-SpikingVTG w/ SFG | 65.31 | 47.48 | 40.35 | 67.30 | **INT-ACC** | ✓ | **0.19** | **1.3mJ** |
| 1-bit (NF)-SpikingVTG w/ ReLU | 65.91 | 47.04 | 40.16 | 67.07 | **INT-ACC** | ✓ | **0.19** | **1.3mJ** |

**Ablation Study:** As demonstrated in Table 5, the inclusion of the SFG mechanism enhances performance compared to the model without SFG. It results in reduced neuronal activity and overall less energy consumption. Thus, we enable SFG for the SpikingVTG variants we discuss next. Without non-local operations, the (NF)-SpikingVTG model achieves competitive performance even compared to other SOTA non-spiking VTG models. Although the 1-bit (NF)-SpikingVTG variant shows a slight reduction in performance, it is highly memory efficient and involves simpler INT-ACC computations, resulting in order of magnitude less energy consumption. Along with energy consumption ($T_c = 10$) Table 5, also highlights the **sparsity** in the SpikingVTG variants particularly in the 1-bit version underscoring considerably **reduced mean neural activity**. For a more hardware friendly model, we train a variant of 1-bit (NF)-SpikingVTG by replacing all GELU layers with ReLU layer [40] and observe minimal degradation in performance. Extensive hyper-parameter study is done in Appendix E.

**Analysis of Energy Efficiency:** We analyze the test-time energy efficiency of SpikingVTG variants compared to a non-spiking transformer-based model with same depth and hidden dimensions. The analysis considers arithmetic operation costs using a 45nm CMOS technology with 32-bit precision, where FP-MAC, FP-ACC, and INT-ACC ops. consume 4.6 pJ, 0.9 pJ, and 0.1 pJ respectively [41]. The energy cost for a non-spiking transformer encoder layer based on total FLOPs used in attention and linear layers, is $E_A = [(3LD^2) + (LD^2 + L^2D) + (2LD^2)] \times 4.6$ pJ. Considering $L = 200$, $D = 1024$, we compute $E_A = 5.98$ mJ. Since, UniVTG has 4 encoder layers, so total energy is $23.92 mJ$.

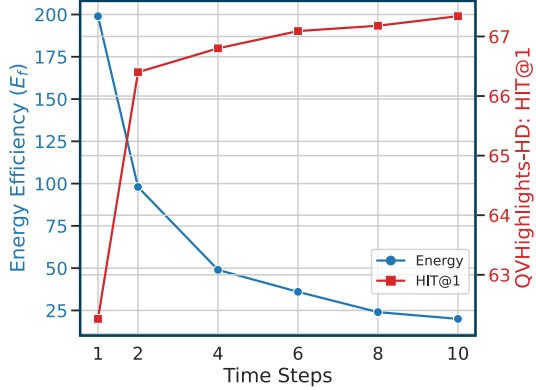

Figure 4: Graph showing tradeoff of energy efficiency ($E_f$) and HIT@1 on QVHighlights-HD for varying time steps.

In contrast, the 1-bit (NF)-SpikingVTG operates over $T_c$ time steps. Its per time-step cost is: $E_{S_t} = [(3 \cdot IFR_{\text{in}} \cdot LD^2) + (IFR_{\text{k}} \cdot LD^2 + IFR_{\text{v}} \cdot L^2D) + (IFR_{\text{attn}} \cdot LD^2) + (IFR_{\text{int.}} \cdot LD^2)] \times 0.1$ pJ, where $IFR_l$ denotes the mean activity of component $l$. For our 1-bit (NF)-SpikingVTG model, which uses INT-ACC operations, we empirically measure: $IFR_{\text{in}} = 0.40$, $IFR_{\text{k}} = 0.18$, $IFR_{\text{v}} = 0.19$, $IFR_{\text{attn}} = 0.03$, and $IFR_{\text{interm.}} = 0.09$, resulting in $E_{S_t} = 0.03$ mJ. With $T_c = 10$, the total spiking energy is $E_S = T_c \cdot E_{S_t} = 0.3$ mJ, yielding an energy efficiency of $E_f = E_A/E_S = 19.93$. Model-wise energy comparison is shown in Table 5.

**Energy-Accuracy Tradeoff:** Unlike conventional non-spiking VTG solutions, SpikingVTG enables a controllable trade-off between performance and energy consumption, as total energy usage scales with the number of operating time steps. In Fig. 4, we demonstrate this performance-energy tradeoff achieved using 1-bit (NF)-SpikingVTG model by running it for different number of time steps. Notably, when running the model for just 2 time steps, SpikingVTG attains a HIT@1 score of 66.4 (compared to the SOTA score of 68.77), while achieving an energy efficiency factor $E_f \approx 100$.

# 6 Conclusions

In this paper we propose SpikingVTG, a bio-inspired computationally efficient solution for VTG tasks in resource constraint environment. We propose a saliency feedback gating mechanism, which leverages the temporal dynamics of the model to improve performance while lowering computational costs by reducing overall neuronal activity across the model. We empirically demonstrate the convergence dynamics of the SpikingVTG model and leverage the formulated steady-state equations to efficiently train our model using implicit differentiation at equilibrium. To improve generalization, we propose a Cos-L2 Representation Matching loss, enabling knowledge transfer from non-spiking VTG models and yielding substantial performance gains. We further optimize the model by removing non-local operations and applying extreme quantization, leading to the 1-bit NF-SpikingVTG. Our framework significantly improves computational efficiency and model compactness, enabling tunable energy-accuracy tradeoffs on low-resource devices. While this work presents the first spiking solution for VTG with competitive performance, there remains room for improvement in fully closing the performance gap. Future research can build upon our framework to further advance spiking models in this domain.

# 7 Acknowledgments

This work was supported in part by the United States Air Force and Defense Advanced Research Projects Agency (DARPA) under Contract No. FA8750-23-C-0519 and the U.S. Army Research Laboratory Cooperative Research Agreement W911NF-17-2-0196. Any opinions, findings and conclusions or recommendations expressed in this material are those of the authors and do not necessarily reflect the Department of Defense or the United States Government.

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

# A  Extended Architecture Overview

The Spiking Transformer layer primarily consists of a spiking multi-head attention (MHA) block, followed by a spiking feedforward network comprising an intermediate layer and an output layer with both inter- and intra-layer communication happening using spikes. Details of the operations in each layer are provided below.

**Spiking Attention Block:** In Spiking MHA, to enable computationally efficient accumulate based operations the input to the attention layer are spikes instead of real-valued data. The spiking attention mechanism [26] is given as follows,

$$Attn(X_s[t], K_s[t], V_s[t]) = \phi(d * Q(X_s[t]) \cdot (K_s[t])^T) \cdot V_s(t) \tag{8}$$

Here, $Q(X_s(t))$ represents the Query, obtained by passing the input spikes $X_s(t)$ at time $t$ through a linear layer ($W_Q$). The spikes for the Key layer ($K_s(t)$) are generated by passing $X_s(t)$ through a linear mapping ($W_K$), followed by an LIF neuron layer. Similarly, we generate spikes for Value. $d$ is a scaling constant. Since the input, key, and value matrices consist of spike trains rather than real-valued data, the primary computations in all matrix multiplications are floating-point accumulation operations rather than floating point multiplicative and accumulative operations. In the (NF)-SpikingVTG variant, as discussed in the paper, we use $\phi$ as the $ReLU$ and scaling operation, significantly reducing the computational overhead compared to employing $\phi$ as the non-local $Softmax$ operation. The output of the attention layer is fed to an LIF neuron, which outputs spikes. The convergence dynamics of the layer at equilibrium is given as, $a_{attn}^* = \sigma(\frac{1}{V_{th}}(Attn(a_x^*, a_k^*, a_v^*) + b_{attn})$, where $a_x$ represents the ASR of the layer used to generate the Query, $a_k$ denotes the ASR of the Key, and $a_v^*$ corresponds to the ASR of the Value. $b_{attn}$ is a bias term.

**Intermediate Layer:** The intermediate layer takes as input the spikes generated from the preceding layer and maps it to an intermediate dimension with a linear layer. The output is then passed through an LIF layer. The convergence dynamics of the layer at equilibrium is given as, $a_{interm.}^* = \sigma(\frac{1}{V_{th}}(act(W_{interm.}a_p^*) + b_{interm.}))$, where $W_{interm.}$ is the linear weight and gelu() is the activation used for the layer. $a_p^*$ is the ASR at equilibrium for the previous layer. $b_{interm.}$ is a bias term. In this paper, we have used explored different choices for $act$, such as $GELU$ and $ReLU$. During inference, all matrix multiplications involve accumulative operations due to the nature of the input.

**Output Layer:** The output layer takes as input the spikes generated from the preceding layers as shown in Fig. 1. The output is then passed through an LIF layer. The convergence dynamics of the layer at equilibrium is given as, $a_{output}^* = \sigma(\frac{1}{V_{th}}(norm(W_{output}a_{interm.}^* + a_p^*) + b_{output}))$, where $W_{output}$ is the linear weight and layer norm is used for normalization. $a_{interm.}^*$ is the ASR at equilibrium for the previous intermediate layer. $b_{output}$ is a bias term. During inference, all matrix multiplications involve accumulative operations due to the nature of the input. In the (NF)-SpikingVTG model we further remove the layer normalization to improve on-chip deployability.

# B  Loss Function Details

As described in the main paper, the total loss over $N$ clips in the training set is defined as $L = \frac{1}{N} \sum_{i=1}^{N} (L_{f_i} + L_{d_i} + L_{c_i})$, where $L_f$ represents the binary cross-entropy loss for the indicator variable $f_i$, $L_d$ combines the smooth L1 loss with the generalized IoU loss [31] for the predicted boundaries, and $L_c$ is an optional loss term incorporating intra- and inter-video contrastive learning [32]. We follow similar loss function construction as previous works on VTG [1, 6]. The loss for fore-ground parameter is given as follows,

$$L_f = -\lambda_f \left[ f_i \log \tilde{f}_i + (1 - f_i) \log(1 - \tilde{f}_i) \right] \tag{9}$$

where, $f_i$ is the true label and $\tilde{f}_i$ is the model prediction. The loss for predicted boundaries is given as follows,

$$L_d = \mathbf{1}_{f_i=1} \left( \lambda_{\text{L1}} L_{\text{SmoothL1}}(\tilde{d}_i, d_i) + \lambda_{\text{iou}} L_{\text{iou}}(\tilde{b}_i, b_i) \right) \tag{10}$$

where, $d_i, b_i$ are the true label and $\tilde{d}_i, \tilde{b}_i$ is the model prediction. $L_c = \lambda_{\text{inter}} L_{\text{inter}} + \lambda_{\text{intra}} L_{\text{intra}}$ is used for inter-video and intra video contrastive learning [6]. For each video $V$, we randomly select a clip $v_i$ with fore-ground indicator = 1 and positive saliency score. Clips from the same video, denoted as $v_j$, with saliency scores $s_j < s_i$ are treated as negative samples. i.e., $A = \{j \mid s_j < s_i, 1 \leq j \leq L_v\}$, and perform intra-video contrastive learning using the loss

$$L_{\text{intra}} = -\log \frac{\exp(\tilde{s}_i/\tau)}{\exp(\tilde{s}_i/\tau) + \sum_{j \in A} \exp(\tilde{s}_j/\tau)} \tag{11}$$

. Furthermore, we treat textual queries from other samples within the batch ($k \in S$) as negative samples, enabling inter-video contrastive learning for cross-sample supervision:

$$L_{\text{inter}} = -\log \frac{\exp(\tilde{s}_i/\tau)}{\sum_{k \in S} \exp(\tilde{s}_i^k/\tau)} \tag{12}$$

, where $S$ is the training batch, $\tilde{s}_i^k = \cos(v_i, M_k)$ and $M_k$ is the sentence representation (Eqn. 2) and $cos$ is cosine similarity.

## C   Dataset Details

**QVHighlights:**  The QVHighlights dataset [1] stands out as the sole dataset providing annotations for both moment retrieval and highlight detection, making it an excellent resource for benchmarking on both the VTG tasks. Comprising 10,148 videos with an average duration of 150 seconds. It features a total of 10,310 queries linked to 18,367 moments, resulting in an average of 1.8 distinct moments per query within each video. The dataset spans a variety of scenarios, including daily vlogs, travel vlogs, and news events.

**Charades-STA:**  The Charades-STA [42] dataset comprises 16,128 indoor videos, each with an average duration of 30.6 seconds. It includes 12,408 query-interval pairs designated for training and 3,720 query-interval pairs reserved for testing.

**TACoS:**  TACoS [43] consists of 127 videos, each averaging 4.78 minutes in length. The dataset is split into 75 videos for training, 27 for validation, and 25 for testing.

**Youtube Highlights:**  YouTube Highlights [44] consists of 433 videos across 6 domains, using the domain names as text queries.

## D   Evaluation Metrics:

For QVHighlights, following previous work [1] we use Recall@1 with IoU thresholds of 0.3, 0.5 and 0.7 and avg. mean average precision (mAP), mAP@0.5 and mAP@0.75 as the evaluation metric for moment retrieval tasks. For highlight detection, we use mAP and HIT@1 [1], where a clip is considered a true positive if it receives a score of "Very Good" [5]. For Charades-STA and TACoS, we employ Recall@1 with IoU thresholds of 0.3, 0.5, and 0.7, along with the mean IoU (mIoU). For Youtube Highlights we use mAP.

## E   Additional Experimental Details

In this subsection, we provide a concise overview of the implementation details and provide additional experimental details. The GPU specifications for the experiments are detailed in the main paper, while the CPU utilized is an AMD Ryzen Threadripper 3960X 24-Core Processor. We have used Python and the PyTorch framework to write the code. The video and textual feature are developed following previous work [1, 6]. We have used the Adam optimizer to train our model. We list the hyper-parameters used in the work in Table 6. We used grid search to find optimal values.

### E.1   Training Stages

Training a multi-modal spiking architecture like SpikingVTG is resource-intensive. To enhance the efficiency of this process and develop computationally efficient variants of our model, we leverage a

| Hyper-parameters | Range | Optimal |
|---|---|---|
| $N$: Encoder Layers | (2-6) | 4 |
| $D$: Hidden Dimension | (768-2048) | 1024 |
| $n_1$: $f$-decoder depth | (1-5) | 3 |
| $k_1$: $f$-decoder kernel size | (3-9) | 3 |
| $n_2$: $d$-decoder depth | (1-5) | 3 |
| $k_2$: $d$-decoder kernel size | (3-9) | 7 |
| $T_{CLRM}$: Timesteps for CLRM | (5-100) | 50 |
| $T_f$: Timesteps for Finetuning | (5-50) | 16 |
| $V_{th}$: Threshold Potential | (0.5 - 2.0) | 1.0 |
| $\gamma$: Leaky-factor | (0.9 - 1.0) | 0.99 |
| $\lambda_f$ :$L_f$ co-efficient | (1 - 20) | 10 |
| $\lambda_{L1}$ :$L_{SmoothL1}$ co-efficient | (1 - 20) | 10 |
| $\lambda_{intra}$ :$L_{intra}$ co-efficient | (0 - 1.0) | 0.05 |
| $\lambda_{inter}$ :$L_{inter}$-co-efficient | (0 - 1.0) | 0.01 |
| $\lambda_{iou}$ :$L_{iou}$ co-efficient | (1 - 20) | 10 |
| $\lambda_{cos}$: Sq. cosine weight (CLRM) | (0 - 1.0) | 0.2 |
| $\lambda_{l_2}$: $L_2$ weight (CLRM) | (0 - 1.0) | 0.8 |
| $lr$: Learning Rate | $(1e^{-5} - 1e^{-6})$ | $8e^{-6}$ |
| $w_d$: weight decay | $(1e^{-5} - 1e^{-3})$ | $1e^{-4}$ |
| Batch Size | (8-64) | 32 |
| Epochs: CLRM | 10-100 | 50 |
| Epochs: Finetuning | 20-200 | 100 |

Table 6: Hyper-parameters of our SpikingVTG model. Optimal values for QVHighlights dataset is also shown.

multi-staged training framework. We utilize a transformer-based non-spiking VTG model (such as UniVTG) to perform CLRM loss optimization. After this initial stage, we fine-tune SpikingVTG using the true labels. Once the base SpikingVTG model is established, we modify its architecture to remove non-local operations and perform extreme quantization, followed by additional fine-tuning to create computationally efficient variants with minimal performance degradation. The resulting computationally efficient, lightweight models are well-suited for deployment on neuromorphic chips, enabling efficient inference.

