# OpenReview forum: "SpikingVTG: A Spiking Detection Transformer for Video Temporal Grounding"
_NeurIPS.cc/2025/Conference — NeurIPS 2025 poster_

### Official Review · Reviewer_k5ce · 2025-07-01

**Clarity:** 2
**Significance:** 3
**Originality:** 2
**Rating:** 4
**Confidence:** 3

**Summary:**

This paper proposes SpikingVTG, a novel SNN-based architecture designed for the Video Temporal Grounding (VTG) task. The method segments a video into fixed-length clips and predicts, for each clip, three values: a binary foreground indicator, a temporal offset pair, and a relevance score.
A key contribution is the Saliency Feedback Gating (SFG) mechanism, which uses intermediate outputs to compute dynamic saliency scores and update video input features over discrete timesteps. The paper also formulates the convergence dynamics of the spiking transformer to steady-state ASR values and utilizes these in the training and inference pipeline.

**Questions:**

Could the authors clarify how convergence to equilibrium is defined and detected during training?
How do the authors ensure the compatibility of textual features and deep-layer video ASRs in the cosine similarity calculation within SFG?
Has the computational cost of SFG (per-timestep cosine similarity) been evaluated on longer video sequences or real-time settings?
Is the SFG module quantized in the 1-bit NF-SpikingVTG model? If not, could its floating-point operations impact the claimed energy efficiency?

**Ethical Concerns:**

["NO or VERY MINOR ethics concerns only"]

**Final Justification:**

I appreciate the authors’ efforts in providing a thorough rebuttal and clarifications. Many of my earlier concerns have been addressed. My overall assessment remains unchanged at Borderline accept.

**Limitations:**

yes

**Quality:**

3

**Strengths And Weaknesses:**

Strengths
This is likely the first attempt to apply spiking neural networks to the VTG domain, introducing a new perspective on low-power temporal reasoning tasks.
The SFG mechanism is biologically inspired and dynamically adjusts input features based on feedback from average spiking rates (ASR), allowing for adaptive and sparse computation.
Weakness
Assumption of Equilibrium: While the paper claims that ASR values and saliency scores converge to equilibrium and provides partial visualizations, it lacks a rigorous empirical or theoretical analysis of the convergence process. A more precise treatment of convergence criteria, dynamics, generalizability, and other related aspects would strengthen the claims made in this section.
Feature Space Compatibility: In the SFG module, the cosine similarity is computed between pooled text representations M and ASRs from deep layers a_i^{Nv[t]}. However, the compatibility between these feature types (from different depths and modalities) is not discussed.
Gating Mechanism Scalability: The computation of cosine similarity per clip and timestep may add overhead, and the potential latency of SFG as a feedback loop in larger-scale settings is not discussed.

---

> ### Author Rebuttal · Authors · 2025-07-31
>
> We sincerely thank the reviewer for their thoughtful and insightful feedback. Below, we provide detailed responses to each of the questions raised.
>
> **Question 1**
> > Could the authors clarify how convergence to equilibrium is defined and detected during training?
>
> **Response**
> Thank you for this question. In our framework, convergence to equilibrium is defined based on the discrete-time dynamics of per neuron average spiking rate (ASR). Below, we provide the overview of a theoretical proof of convergence following [10]. For simplicity we show the proof when all layers are linear and using Integrate-Fire neurons with binary spikes  \([0, 1]\) and define the average spiking rate (ASR) over \(t\) steps as:  $a_l[t] = \frac{1}{t} \sum_{\tau=1}^{t} s_l[\tau],$ which is bounded within \([0, 1]\) because there can be at most one spike per timestep.. For the $\(l\)$-th layer, the membrane potential update is given by:
>
> $u_l[t + 1] = u_l[t] + W s_{l-1}[t] + b - V_{\text{th}}\ s_l[t + 1],$
> where $\(s_{l-1}[t]\)$ is the spike train from the previous layer and $\(V_{\text{th}}\)$ is the firing threshold.
>
>
> Following [10], we can decompose the membrane potential into negative and exceeded positive terms, $u\_i[t] = u\_i^{-}[t] + u\_i^{+}[t]$. If the instantaneous input is negative, the neuron will not emit a spike, and this inhibitory contribution remains in the membrane potential as the **negative term** $u\_i^{-}[t]$. The **positive term** $u\_i^{+}[t]$ represents the excitatory drive, typically bounded between $0$ and $V\_{\text{th}}$. In our convergence analysis, we assume $u\_i^{+}[t]$ is bounded by a constant. Thus, with a linear feedback ($F$) the  ASR of the first layer and subsequent layers becomes:
>
> $a\_1[t+1] = \sigma\\left(\frac{1}{V\_{\text{th}}} \left(\frac{t}{t+1} F a\_N[t] + W\_1 x[t] + b\_1\right)\right) - \frac{1}{V\_{\text{th}}} \cdot \frac{u\_1^{+}[t+1]}{t+1}$,
> $a\_{l+1}[t+1] = \sigma\\left(\frac{1}{V\_{\text{th}}} \left(W\_{l+1} a\_l[t+1] + b\_{l+1}\right)\right) - \frac{1}{V\_{\text{th}}} \cdot \frac{u\_{l+1}^{+}[t+1]}{t+1}$, for $l = 1, \dots, N-1$.
>
>
> where, $\sigma(x) = 1$ if $x > 1$, $\sigma(x) = x$ if $0 \le x \le 1$, and $\sigma(x) = 0$ if $x < 0$.
>
>
> If the average inputs converge $(x^{\*})$, the exceeded terms are bounded $(\(|u_l^{+}[t]| \le c\))$, and the product of spectral norms satisfies  $\|F\|\_2 \|W\_1\|\_2 \|W\_2\|\_2 \cdots \|W\_N\|\_2 \le \gamma V\_{\text{th}}^{N}$ with $\gamma < 1$, then by iterative contraction we have:
> $\[
> \|a_1[t+1] - a_1[t]\| \le \gamma \|a_1[t] - a_1[t-1]\| + \epsilon (1-\gamma)/2,
> \]$
> for sufficiently large \(t\). By Cauchy’s convergence test, $\(a_1[t] \to a_1^*\)$, where:
>
> $a\_1^\* = f\_1(f\_N \circ \cdots \circ f\_2(a\_1^\*), x^\*)$, and inductively $a\_{l+1}^\* = f\_{l+1}(a\_l^\*)$ for all $l$. where $f\_1(a, x) = \sigma\\left(\frac{1}{V\_{\text{th}}}(F\ a + W\_1 x + b\_1)\right)$ and
>
> $f\_{l+1}(a) = \sigma\left(\frac{1}{V\_{\text{th}}}(W\_{l+1} a + b\_{l+1})\right)$.
>
> We will add this proof to the appendix with more details.
>
>
> **Empirical Convergence**}: Empirically, we observe that the ASR across all layers converges within a small number of time steps, typically $T_c \in [5,10]$. As shown in Fig.~3a of the main paper, the layer-wise ASR stabilizes rapidly. Consequently, we treat the ASR at the final simulation step, $a[T_{c}]$, as an approximate equilibrium value $a^*$. During training, we leverage this converged ASR to compute gradients via implicit differentiation at equilibrium (Section 4.5).
>
> We hope this response clarifies both the theoretical foundations and empirical detection strategy for equilibrium convergence in our framework.
>
> **Question 2**
>
> > **How do the authors ensure the compatibility of textual features and deep-layer video ASRs in the cosine similarity calculation within SFG?**
>
> **Response**
>
> We appreciate the reviewer’s thoughtful question. In the **Saliency Feedback Gating (SFG)** mechanism, the cosine similarity is computed between the **Average Spiking Rate (ASR)** of the final spiking transformer layer corresponding to each video segment
>   $\( a^{N_v}_i[t] \in {R}^D \)$, and the **sentence-level textual embedding** $\( M \in {R}^D \)$,
>
> as defined in Eq. (2) of the paper: $F^v_s[i,t] = \cos(a^{N_v}_i[t], M) = \frac{a^{N_v}_i[t] \cdot M}{\|a^{N_v}_i[t]\|_2 \|M\|_2}.$
>
> To ensure compatibility of these two modalities in the cosine similarity computation, we incorporate the following design choices:
> Both $\( a^{N_v}_i[t] \)$ and $\( M \)$ reside in a latent space of dimensionality $\( D \)$. The query representation $\( Q \in \mathbb{R}^{L_q \times D} \)$ is projected to a global sentence embedding ($\mathbb{R}^{D}$) via attention-based pooling:$\[M = Q^\top \text{Softmax}(Q W_p),\]$ where $\( W_p \in \mathbb{R}^{D \times 1} \)$ is a *trainable* projection matrix. This learnable mapping helps align the textual representation with the spatiotemporal semantics of the spiking video features, thereby enabling cross-modal compatibility.
>
> **Empirical validation:**
> As shown in **Fig. 2b** of the main paper, clips semantically related to the textual query exhibit higher saliency scores $\( F^v_s \)$, demonstrating that the representations are well-aligned. The effectiveness of this design is further reflected in **improved performance** and **reduced neural activity** with the SFG mechanism (see Table 5 and Fig. 3).
>
> **Question 3**
> > Has the computational cost of SFG (per-timestep cosine similarity) been evaluated on longer video sequences or real-time settings? Is the SFG module quantized in the 1-bit NF-SpikingVTG model? If not, could its floating-point operations impact the claimed energy efficiency?
>
> **Response**
>
> **Computational Cost of SFG:**
>  The SFG module computes per-clip saliency scores via cosine similarity between the output layer ASR ($a^{N_v}_i[t] \in \mathbb{R}^D$) and the global sentence embedding $\( M \in \mathbb{R}^D \)$.  This results in a total computational complexity of  $\[\mathcal{O}(L_v \cdot D)\]$  which is *linear* in the number of video segments $\( L_v \)$. In contrast, the spiking attention mechanism in the transformer core has quadratic complexity  $\[\mathcal{O}(L^2 \cdot D + L \cdot D^2)\]$  where $\( L = L_v + L_q \)$. Thus, *even for long video sequences*, the computational cost of SFG is negligible compared to the spiking transformer core.  Moreover, the operations involved in SFG consist of dot products—element-wise in nature—and are computationally lightweight.
>
> **SFG in 1-bit (NF)-SpikingVTG:**  Yes, in the 1-bit (NF)-SpikingVTG model, the SFG module is fully compatible with the quantized architecture. No softmax operation is used in SFG. Instead, we apply ReLU followed by scaling as discussed in the paper avoiding non-local operations.  The projection matrix $W_p$ used to compute the sentence embedding is also quantized to 1-bit.
>
> **Energy Considerations of SFG**  While cosine similarity involves floating-point arithmetic (element-wise multiplication and summation), it does **not** require floating point matrix multiplications.  Therefore, its energy footprint is **significantly lower** than that of transformer attention layers.  For example, for the 1-bit NF SpikingVTG, considering cost of arithmetic operations on 45nm CMOS, while the transformer core consumes 1.3mJ the SFG mechanism only consumes ~0.01mJ only (Considering a video of 400 secs).
>
> **Thank you for your insightful review and comments. In light of our clarifications, we respectfully request the reviewer to reconsider their rating. We are happy to address any additional questions or comments.**

---

> > ### Author Response · Authors · 2025-08-08
> >
> > Thank you for your positive review and thoughtful questions. As the discussion period concludes today, we sincerely hope you might reconsider your rating in light of the clarifications provided in our rebuttal.

---

> > > ### Author Response · Authors · 2025-08-09
> > >
> > > We sincerely appreciate you taking the time to carefully review our rebuttal. We would be grateful to know if our responses have helped in clarifying our work.

---

### Official Review · Reviewer_UKZa · 2025-07-02

**Clarity:** 3
**Significance:** 2
**Originality:** 2
**Rating:** 4
**Confidence:** 3

**Summary:**

The paper introduces SpikingVTG, a novel Spiking Neural Network based model designed for Video Temporal Grounding tasks, which aims to identify precise temporal segments in a video corresponding to natural language queries.
The SpikingVTG architecture consists of three key components: a spiking Transformer core, a Saliency Feedback Gating mechanism, and a spiking decoder for output generation. The spiking Transformer core utilizes Leaky Integrate-and-Fire neurons as its fundamental computational units and replaces the traditional floating-point multiply-accumulate operations in standard Transformers with more efficient floating-point accumulate-only operations, enabling it to capture temporal and cross-modal dependencies effectively.

**Questions:**

Please refer to Weaknesses.

**Ethical Concerns:**

["NO or VERY MINOR ethics concerns only"]

**Final Justification:**

The issues about fusion method of video and language, individual contributions of each component, comparison with other lightweight VTG architectures and the deployment issues were mostly solved. Although there are some minor issues, the overall impact of these issues is relatively small and does not significantly affect the overall justification of this paper. Therefore, I recommend a borderline accept.

**Limitations:**

Yes

**Quality:**

3

**Strengths And Weaknesses:**

Strengths
* The paper introduces a novel application of Spiking Neural Networks (SNNs) to the domain of Video Temporal Grounding (VTG), which has been traditionally dominated by conventional neural networks. This innovation leverages the computational efficiency and sparsity of SNNs, offering a bio-plausible and energy-efficient alternative.
* By exploiting the inherent sparsity and accumulation-based computation of SNNs, the model achieves substantial reductions in computational overhead and energy consumption. The introduction of normalization-free and 1-bit quantized variants further underscores the model’s suitability for deployment on resource-constrained edge devices, directly addressing a critical practical challenge in VTG.
* The authors provide extensive experimental results, including ablation studies and comparisons with state-of-the-art non-spiking models. This comprehensive evaluation supports the claims made in the paper and provides insights into the model's behavior.

Weaknesses
* The paper does not provide in-depth discussion or ablation on how the spiking framework handles the fusion of video and language modalities, which is a non-trivial challenge in VTG. The reader is left to assume that the spiking transformer core adequately captures cross-modal dependencies, but more detailed analysis or visualization would strengthen the claim.
* Although the SFG mechanism and other architectural innovations are well-motivated, the paper lacks thorough ablation studies to quantify the individual contributions of each component (e.g., normalization-free design, Cos-L2 transfer). This makes it difficult to assess which parts of the design are most critical to performance.
* The experimental section primarily compares SpikingVTG to non-spiking transformer models. It does not benchmark against other lightweight, quantized, or pruned VTG architectures, which would be necessary to convincingly demonstrate the unique advantages of SNNs in this context.
* The model is positioned as suitable for edge deployment, but the paper does not report real-time inference latency or throughput on actual hardware. Without these metrics, the practical viability for time-sensitive applications remains somewhat speculative.
* While the paper claims significant energy efficiency, the methodology for measuring energy consumption and the hardware context are not described in sufficient detail. This lack of transparency may limit reproducibility and make it difficult to directly compare with other efficient VTG models.

---

> ### Author Rebuttal · Authors · 2025-07-31
>
> We thank the reviewer for their insightful feedback. Below, we provide our responses to each of the comments.
>
> **Weakness 1**
> > The paper does not provide in-depth discussion or ablation on how the spiking framework handles the fusion of video and language modalities, which is a non-trivial challenge in VTG. The reader is left to assume that the spiking transformer core adequately captures cross-modal dependencies, but more detailed analysis or visualization would strengthen the claim.
>
> **Response**
>
> We thank the reviewer for raising this point. In SpikingVTG, cross-modal fusion is achieved through joint spiking attention over concatenated video and query tokens. Given the spike-based input
> $X[t] = [V_s[t]; Q_s[t]] \in \mathbb{R}^{L \times D}$, attention weights are computed following Eqn. 8 (in paper) as:
>
> $\alpha_{ij}[t] = \varphi\left(s * Q(X_i[t]) \cdot (K_j[t])^\top\right),$
>
> enabling interaction between video and query tokens $(\(i \in V, j \in Q\))$. This joint attention mechanism allows the model to learn cross-modal dependencies efficiently. Furthermore, the Saliency Feedback Gating (SFG) mechanism applies to the input video tokens. The gated video input is:
>
> $\bar{V}[t+1] = V[t]  * \bar{F}_s^v[t], \quad \text{with } \bar{F}_s^v[t] \in [0, 1]^{L_v},$ where $F^v_s[i,t] = \cos(a^{N_v}_i[t], M) = \frac{a^{N_v}_i[t] \cdot M}{\|a^{N_v}_i[t]\|_2 \|M\|_2}.$
>
> To ensure compatibility of these two modalities in the cosine similarity computation, we incorporate the following design choices:
> Both $\( a^{N_v}_i[t] \)$ and $\( M \)$ reside in a latent space of dimensionality $\( D \)$. The query representation $\( Q \in \mathbb{R}^{L_q \times D} \)$ is projected to a global sentence embedding ($\mathbb{R}^{D}$) via attention-based pooling:$\[M = Q^\top \text{Softmax}(Q W_p),\]$ where $\( W_p \in \mathbb{R}^{D \times 1} \)$ is a *trainable* projection matrix. This learnable mapping helps align the textual representation with the spatiotemporal semantics of the spiking video features, thereby enabling cross-modal compatibility. In the paper, Figure 2b visualizes the learned saliency scores, showing higher values near target clip.
>
> **Weakness 2**
> > Although the SFG mechanism and other architectural innovations are well-motivated, the paper lacks thorough ablation studies to quantify the individual contributions of each component (e.g., normalization-free design, Cos-L2 transfer). This makes it difficult to assess which parts of the design are most critical to performance.
>
> **Response**
>
> We thank the reviewer for this helpful suggestion. Below, we present additional ablations that quantify the contributions of (i) Cos-L2 Representation Matching (CLRM), (ii) Normalization-Free (NF) design, and (iii) 1-bit quantization on QVHighlights.
>
> (1) Effect of Cos-L2 Representation Matching (CLRM)
>
> | Method                 | @0.5      | @0.7      | mAP       | HIT\@1    |
> | ---------------------- | --------- | --------- | --------- | --------- |
> | SpikingVTG w/o CLRM    | 60.12     | 39.68     | 36.23     | 62.49     |
> | **SpikingVTG w/ CLRM** | **67.58** | **50.82** | **40.81** | **68.64** |
>
> CLRM significantly improves performance by aligning the spiking model’s internal states and attention maps with a pretrained non-spiking teacher, enabling strong generalization without costly pretraining.
>
> (2) Effect of NF and 1-bit Quantization on Accuracy and Efficiency
>
> | Method                       | @0.5      | @0.7      | mAP       | HIT\@1    | Energy (mJ) | Neural Activity |
> | ---------------------------- | --------- | --------- | --------- | --------- | ----------- | --------------- |
> | SpikingVTG w/ SFG       | 67.58 | 50.82 | 40.81     | 68.64 | 13.8        | 0.34           |
> | (NF)-SpikingVTG w/ SFG       | 66.59     | 48.31     | 40.61 | 67.73     | 10.1        | 0.25           |
> | 1-bit (NF)-SpikingVTG w/ SFG | 65.31     | 47.48     | 40.35     | 67.30     | 1.3     | 0.19       |
>
> Thus, removing non-local operations and enabling extreme quantization results in reduced computational overhead with minimal reduction in model performance.
>
> **Weakness 3**
> > The experimental section primarily compares SpikingVTG to non-spiking transformer models. It does not benchmark against other lightweight, quantized, or pruned VTG architectures, which would be necessary to convincingly demonstrate the unique advantages of SNNs in this context.
>
> **Response**
>
> We thank the reviewer for this valuable suggestion. Our primary objective is to present SpikingVTG as the first spiking neural architecture for Video Temporal Grounding (VTG). To the best of our knowledge, this is the first work to investigate VTG from a neuromorphic, spike-driven computation perspective. In the absence of prior spiking VTG frameworks, we benchmark SpikingVTG against strong non-spiking, transformer-based VTG models, which currently represent the state of the art on standard datasets such as QVHighlights, Charades-STA, and TACoS. In addition, we have considered and positioned our method in relation to recent efficient VTG approaches, as discussed below:
>
> **SpikeMba (Li et al., 2024)**: This hybrid model integrates spiking neurons into a Mamba-based architecture. However, its backbone is based on the Mamba architecture and remains fundamentally non-spiking and relies heavily on dense floating-point matrix multiplications. In contrast, 1-Bit NF SpikingVTG is a fully spiking architecture—including transformer, decoder, and feedback pathways—and avoids matrix multiplications altogether during inference by relying on integer accumulation (INT-ACC) operations, making it better suited for deployment on neuromorphic hardware (as discussed in the next segment).
>
> **$R^{2}$-Tuning (ECCV 2024)**: $R^{2}$-Tuning improves efficiency by exploring image-to-video transfer learning via learning  lightweight modules with reduced parameter count. While efficient in terms of model size, it still relies on full-precision dense computation during inference. SpikingVTG is fundamentally different in that it introduces a dynamic, event-driven computation model inspired by biological neurons and achieves significant energy savings without compromising competitive performance.
>
> We will discuss this in more details in the final version of the paper.
>
>
> **Weakness 4,5**
> > The model is positioned as suitable for edge deployment, but the paper does not report real-time inference latency or throughput on actual hardware. Without these metrics, the practical viability for time-sensitive applications remains somewhat speculative. While the paper claims significant energy efficiency, the methodology for measuring energy consumption and the hardware context are not described in sufficient detail. This lack of transparency may limit reproducibility and make it difficult to directly compare with other efficient VTG models.
>
> **Response**
>
> We thank the reviewer for pointing this out. In order to validate the feasibility of on-chip deployment, we have implemented our 1-bit (NF)-SpikingVTG to be compatible with Intel Loihi 2 using the Intel LAVA framework. The model is architecturally aligned with Loihi 2’s hardware capabilities, being fully matmul-free (replacing floating-point MACs with integer accumulation operations), removing all non-local normalization layers such as softmax and layer norm. We implement our ternary spiking neurons using **lava.proc.lif.models.PyTernLifModelFixed** and the 1-bit weights are implemented as a custom **ProcessModel** similar to the BitLinear implementation done by [1].
>
> Loihi 2 supports two execution modes: **pipelined** mode for full-input processing and **fall-through** mode for autoregressive token generation. For VTG tasks, we use pipelined mode for offline batch settings (e.g., full video moment retrieval) and fall-through mode for real-time highlight detection (clip-by-clip processing). While [1] implemented a 12-layer matmul-free transformer (370M parameters), our proposed model has only 4 layers with approximately $10\times$ fewer parameters, further reducing memory and compute requirements.
>
> In fall-through mode that is with realtime processing, reported Loihi 2 benchmarks for a matmul-free transformer akin 1-bit NF SpikingVTG, demonstrate a throughput of 41.5 tokens/sec compared to 14.3 tokens/sec for an iso-param conventional transformer on an NVIDIA Jetson GPU (around $3 \times$ speedup). The energy cost per token when we process the entire sequence is 3.7 mJ/token on Loihi 2 versus 11.7 mJ/token on the Jetson GPU, i.e. $>3 \times$ less.
>
> Apart from leveraging computationally efficient accumulation based operations, these improvements stem from Loihi 2’s on-core storage of weights and recurrent states in local SRAM, which minimizes memory movement and reduces both latency and energy consumption.
>
> References:
>
> [1] Abreu, Steven, Sumit Bam Shrestha, Rui-Jie Zhu, and Jason Eshraghian. "Neuromorphic Principles for Efficient Large Language Models on Intel Loihi 2." arXiv preprint arXiv:2503.18002 (2025).
>
> **We thank the reviewer for their thorough evaluation and valuable feedback. In light of the clarifications provided, we respectfully request the reviewer to reconsider their rating. We are happy to address any additional questions or comments.**

---

> > ### Comment · Reviewer_UKZa · 2025-08-08
> >
> > I appreciate the detailed response provided by the authors.
> > The reply addresses most of my concerns, and I will update my rating positively to reflect this.

---

> > > ### Author Response · Authors · 2025-08-08
> > >
> > > We sincerely appreciate your careful review of our rebuttal and your decision to increase the score. Thank you for your support.

---

### Official Review · Reviewer_Y4Qk · 2025-07-03

**Clarity:** 2
**Significance:** 3
**Originality:** 3
**Rating:** 4
**Confidence:** 3

**Summary:**

The paper proposes SpikingVTG, a spiking neural network based architecture for video temporal grounding tasks. Unlike conventional transformers, it uses spike-based, event-driven computation for higher energy efficiency. Key innovations include a Saliency Feedback Gating mechanism to focus on relevant video clips, a knowledge transfer method (CLRM) from non-spiking models, and quantized, normalization-free variants for edge deployment. The results on the benchmarks shows good performance at low cost.

**Questions:**

1. seems the model accuracy depends on Cos-L2 Representation Matching with a heavyweight teacher; without CLRM the performance drops about 3%on qvh. How important is the teacher model, what if we use a smaller teacher, is the the SNN architecture itself can learn strong multimodal representations or it has to rely on the knowledge distillation?
2. The SFG mechanism produces saliency scores based on average spiking rates, but there is no visualize or explain why certain clips are gated.  Please further explain if certain behaviors are explainable or the behavior of SFG are purely empirical.

**Ethical Concerns:**

["NO or VERY MINOR ethics concerns only"]

**Final Justification:**

i carefully read the rebuttal and other reviews and decided to keep my rating.
Please make sure to include the rebuttal numbers and contents to the final paper.

**Limitations:**

Yes

**Quality:**

3

**Strengths And Weaknesses:**

Strength:
1. The paper is intuitive and driven by biologically motivated. The Saliency Feedback Gating (SFG) re-uses the network’s own spiking activity as a guiding signal, boosting meaningful spikes in clips.
2. The 1-bit version fo the model is impressive with only 11MB  in size and reduces one-pass energy to 1.3 mJ, this has true potential to be deployed to the edge devices.

Weaknesses
1. the proposed method, to match the performance, relies on a heavy teacher models. This multi-stage pipeline (teacher > CLRM > fine-tune) adds additional computes and makes it not clear if the proposed architecture is effective or the distillation part.
2. The performance of proposed model Still lags behind the SOTA. I would encourage the authors to provide more in-depth analysis of the errors, and the limitation of current models.
3. Seems the efficiency claims come from FLOP-based calculations with cost model; this is not very convincing, the author could simply use GPU energy consumption to estimate the efficiency.

---

> ### Author Rebuttal · Authors · 2025-07-31
>
> Thank you for your detailed and valuable feedback on our paper. In this rebuttal, we address your comments.
>
> **Question 1**
> > seems the model accuracy depends on Cos-L2 Representation Matching with a heavyweight teacher; without CLRM the performance drops about 3%on qvh. How important is the teacher model, what if we use a smaller teacher, is the the SNN architecture itself can learn strong multimodal representations or it has to rely on the knowledge distillation?
>
> **Response**
>
> Thank you for this insightful question. We would first like to emphasize that even without incorporating Cos-L2 Representation Matching, our model achieves competitive performance (as highlighted below & Table 1 in paper) compared to the non-pretrained transformer-based VTG baseline, UniVTG.
>
> **Table 1:** Ablation study of the effect of CLRM on SpikingVTG evaluated on the evaluation set of QVHighlights.
>
> | Method                  | QVHighlights-MR @0.5 | QVHighlights-MR @0.7 | mAP@avg | mAP   | HIT@1 |
> |-------------------------|----------------------|----------------------|---------|-------|-------|
> | UniVTG              | 59.74                | 40.90                | 36.13   | 38.83 | 61.81 |
> | SpikingVTG w/o CLRM     | 60.12                | 39.68                | 36.23   | 38.84 | 62.49 |
>
> To achieve improved performance, transformer-based VTG models such as UniVTG typically rely on extensive pre-training on large-scale datasets (e.g., Ego4D, VideoCC). However, pre-training a spiking model from scratch is substantially more computationally expensive. To address this, we employ Cos-L2 Representation Matching (CLRM), which enables direct knowledge transfer from a pre-trained UniVTG to the downstream task—eliminating the need for costly large-scale pre-training. To demonstrate that SpikingVTG is fully capable of learning strong multimodal representations on its own, we also pre-trained it on Ego4D and VideoCC. Achieving similar performance (on QVHighlights), however, required approximately 45 hours of wall-clock time for full pre-training, compared to only ~4 hours for knowledge distillation via CLRM (50 epochs, each epoch ~300 sec). This highlights CLRM as an efficient and compute-friendly solution for rapidly training SpikingVTG with significantly reduced resource requirements.
>
> **Question 2**
> > The SFG mechanism produces saliency scores based on average spiking rates, but there is no visualize or explain why certain clips are gated. Please further explain if certain behaviors are explainable or the behavior of SFG are purely empirical.
>
> **Response**
> Thank you for this excellent question. The SFG mechanism computes saliency scores by measuring the cosine similarity between the final-layer ASR representation of each video clip and the textual query embedding (Eq. 2). These similarity scores are then min–max normalized to lie within $[0, 1]$, where **higher values indicate greater relevance to the query**.
>
> Figure  2b provides a visualization of SFG’s behavior. In this example, the ground-truth highlight corresponds to clip index 28. We observe that clips temporally close to index 28 exhibit higher saliency scores (warmer colors, more red), while clips further away from the relevant segment have lower scores (cooler colors, more blue). This pattern is consistent across examples and indicates that the SFG mechanism tends to assign higher weights to clips that are semantically and temporally aligned with the query.
>
> Thus, the per-clip scores $\(F_s\)$ are interpretable as a relevance measure between the query and video content. The visualization confirms that SFG is able to focus attention on query-relevant regions in a manner consistent with the ground truth.
>
> **Weakness 1**
> >the proposed method, to match the performance, relies on a heavy teacher models. This multi-stage pipeline (teacher > CLRM > fine-tune) adds additional computes and makes it not clear if the proposed architecture is effective or the distillation part.
>
> **Response**
>
>  We have responded to this in the response to **Question 1**.
>
> **Weakness 2**
> >The performance of proposed model Still lags behind the SOTA. I would encourage the authors to provide more in-depth analysis of the errors, and the limitation of current models.
>
> **Response**
>
> Thank you for highlighting this point. While SpikingVTG is, to the best of our knowledge, the first fully spiking solution for VTG tasks, we acknowledge that there remains room for improvement toward matching the best-performing full-precision models. One key factor contributing to the performance gap is the quantization of activations to ternary spike states `{−1, 0, 1}`, which inevitably introduces information loss and reduces representational precision.  Future work can address this by exploring graded or multi-level spiking activations (supported in neuromorphic hardware such as Loihi 2), which can improve representational capacity while retaining the energy efficiency of spike-based computation.  We will include a detailed discussion of these limitations and potential improvements in the revised manuscript.
>
> **Weakness 3**
> > Seems the efficiency claims come from FLOP-based calculations with cost model; this is not very convincing, the author could simply use GPU energy consumption to estimate the efficiency.
>
> **Response**
>
> In our paper we provided a FLOP based estimate following energy consumption of MUL, ACC, operations on 45nm CMOS. In order to validate the feasibility of on-chip deployment, we have implemented our 1-bit (NF)-SpikingVTG to be compatible with Intel Loihi 2 using the Intel LAVA framework. The model is architecturally aligned with Loihi 2’s hardware capabilities, being fully matmul-free (replacing floating-point MACs with integer accumulation operations), removing all non-local normalization layers such as softmax and layer norm. We implement our ternary spiking neurons using **lava.proc.lif.models.PyTernLifModelFixed** and the 1-bit weights are implemented as a custom **ProcessModel** similar to the BitLinear implementation done by [1].
>
> Loihi 2 supports two execution modes: **pipelined** mode for full-input processing and **fall-through** mode for autoregressive token generation. For VTG tasks, we use pipelined mode for offline batch settings (e.g., full video moment retrieval) and fall-through mode for real-time highlight detection (clip-by-clip processing). While [1] implemented a 12-layer matmul-free transformer (370M parameters), our proposed model has only 4 layers with approximately $10\times$ fewer parameters, further reducing memory and compute requirements.
>
> In fall-through mode that is with realtime processing, reported Loihi 2 benchmarks for a matmul-free transformer akin 1-bit NF SpikingVTG, demonstrate a throughput of 41.5 tokens/sec compared to 14.3 tokens/sec for an iso-param conventional transformer on an NVIDIA Jetson GPU (around $3 \times$ speedup). The energy cost per token when we process the entire sequence is 3.7 mJ/token on Loihi 2 versus 11.7 mJ/token on the Jetson GPU, i.e. $>3 \times$ less.
>
> Apart from leveraging computationally efficient accumulation based operations, these improvements stem from Loihi 2’s on-core storage of weights and recurrent states in local SRAM, which minimizes memory movement and reduces both latency and energy consumption.
>
> References:
>
> [1] Abreu, Steven, Sumit Bam Shrestha, Rui-Jie Zhu, and Jason Eshraghian. "Neuromorphic Principles for Efficient Large Language Models on Intel Loihi 2." arXiv preprint arXiv:2503.18002 (2025).
>
> **Thank you for your help in improving our paper. In light of our clarifications, we respectfully request the reviewer to reconsider their rating. We are happy to address any additional questions or comments.**

---

> > ### Comment · Reviewer_Y4Qk · 2025-08-08
> >
> > Thanks for the additional information.
> > I carefully read the rebuttal and other reviews, and would like to keep my rating.
> > There are many details included in the rebuttal, please add them to the final paper so that its a complete study.

---

> > > ### Author Response · Authors · 2025-08-08
> > >
> > > We are grateful for your encouraging feedback and constructive suggestions that have helped enhance our work.

---

### Official Review · Reviewer_vi5H · 2025-07-03

**Clarity:** 2
**Significance:** 2
**Originality:** 3
**Rating:** 4
**Confidence:** 3

**Summary:**

This paper proposes SpikingVTG, a novel spiking neural network-based detection transformer designed for Video Temporal Grounding (VTG). The model features a bio-inspired, energy-efficient architecture comprising a spiking transformer core, a Saliency Feedback Gating (SFG) mechanism for relevance-aware filtering, and a spiking decoder for output generation. To enhance generalization, the authors introduce a Cos-L2 Representation Matching (CLRM) loss for knowledge transfer from pre-trained non-spiking models. Furthermore, they develop normalization-free and 1-bit quantized variants for deployment on resource-constrained edge devices. Experimental results across four VTG benchmarks (QVHighlights, Charades-STA, TACoS, and YouTube Highlights) demonstrate competitive performance and notable energy savings, establishing a solid foundation for future research on spiking-based VTG systems.

**Questions:**

The traditional neural networks like CNN, transformer can be accelerated by hardware like GPUs with CUDA. Is their any matured and popular hardware and hardware acceleration libs to support Spiking Neural Networks (SNNs)?

**Ethical Concerns:**

["NO or VERY MINOR ethics concerns only"]

**Final Justification:**

The authors have addressed many of my concerns, I'd happy to lift the score.

**Limitations:**

yes

**Quality:**

2

**Strengths And Weaknesses:**

**Strengths**

The paper introduces the first spiking transformer architecture for VTG, replacing conventional dense matrix multiplications with event-driven, accumulation-based operations using leaky integrate-and-fire neurons. This design aligns well with neuromorphic computing principles and targets the high energy cost of traditional VTG models.
The proposed Saliency Feedback Gating (SFG) mechanism enables dynamic focus on query-relevant video segments using spike-based saliency estimation. Combined with the use of implicit differentiation for BPTT-free training, the model achieves both efficiency and scalability.
The experimental evaluation is comprehensive, spanning four VTG datasets with multiple model variants (including normalization-free and 1-bit versions). The authors conduct thorough ablation studies and analyze the energy-accuracy tradeoff, enhancing the paper's empirical depth and practical relevance.

**Weaknesses**

Despite its architectural efficiency, the proposed model underperforms compared to state-of-the-art pretrained non-spiking VTG models (e.g., UniVTG), especially at higher IoU thresholds. This may limit its adoption in applications requiring high localization precision.
The SFG module introduces additional floating-point operations, but its computational overhead is not fully quantified or compared against the energy gains, making its efficiency claims less conclusive.
While the authors assert that the model is "well-suited for edge-based deployment," no experiments are conducted on actual edge or neuromorphic hardware. This lack of validation undermines the strength of the deployment claim.
All energy efficiency results are estimated via theoretical cost models. The absence of real-device power profiling or runtime measurements limits the practical credibility of the claimed energy savings.

---

> ### Author Rebuttal · Authors · 2025-07-31
>
> We sincerely thank the reviewer for the thoughtful and detailed feedback. We address each concern below and clarify our contributions with additional evidence and improvements for the final version.
>
> **Question 1**
> > The traditional neural networks like CNN, transformer can be accelerated by hardware like GPUs with CUDA. Is their any matured and popular hardware and hardware acceleration libs to support Spiking Neural Networks (SNNs)?
>
> **Response**
>
> Thank you for this insightful question. Yes, mature hardware platforms and toolchains for SNNs are emerging. **Intel's Loihi 2** supports asynchronous event-driven computation and spiking neuron models, making it well-suited for models like our 1-bit NF-SpikingVTG. Other neuromorphic platforms include IBM's TrueNorth, SpiNNaker, and Akida’s BrainChip. Software Librarires such as NxSDK, SpikingJelly, and LAVA enable SNN training and deployment on these platforms.
>
> Importantly, matmul-free transformer architectures similar to ours have been deployed on Loihi 2 [1], demonstrating improved energy efficiency than deploying transformers on edge-based GPUs, confirming the practicality and efficiency of such models for edge deployment. In response to Weakness 3, we have also discussed methodology to deploy our proposed SpikingVTG model on the Intel Loihi 2 using the Intel Lava framework.
>
> **Weakness 1**
> > Despite its architectural efficiency, the proposed model underperforms compared to state-of-the-art pretrained non-spiking VTG models (e.g., UniVTG), especially at higher IoU thresholds. This may limit its adoption in applications requiring high localization precision.
>
> **Response**
> > We thank the reviewer for this valuable feedback. While our focus is not on directly surpassing full-precision non-spiking models like UniVTG, SpikingVTG aims to offer a first-of-its-kind neuromorphic, computationally-efficient alternative for video temporal grounding in resource-constrained settings. It achieves competitive performance on both QVHighlights (e.g., HD @0.7: 50.82 vs. 52.65) and Charades-STA (e.g., @0.7: 37.16 vs. 38.55), while the 1-bit (NF)-SpikingVTG variant sustains strong performance (47.48 and 36.10, respectively) at over 18× lower energy cost (Table 5).
> A key strength of our model is its tunable energy-accuracy tradeoff (Fig. 4), enabled by varying the number of inference time steps. For example, with just 2 time steps, SpikingVTG achieves HIT@1 = 66.4 on QVHighlights with ~100× greater energy efficiency. This level of test-time adaptability is not possible with conventional VTG architectures, making SpikingVTG particularly well-suited for edge and low-power deployments.
>
> **Weakness 2**
> > The SFG module introduces additional floating-point operations, but its computational overhead is not fully quantified or compared against the energy gains, making its efficiency claims less conclusive.
>
> **Response**
>
> **Computational Cost of SFG:**
>  The SFG module computes per-clip saliency scores via cosine similarity between the output layer ASR $\( a^{N_v}_i[t] \in \mathbb{R}^D \)$ and the global textual embedding $\( M \in \mathbb{R}^D \)$.  This results in a total computational complexity of  $\[\mathcal{O}(L_v \cdot D)\]$  which is *linear* in the number of video segments $\( L_v \)$. In contrast, the spiking attention mechanism in the transformer core has quadratic complexity  $\[\mathcal{O}(L^2 \cdot D + L \cdot D^2)\]$  where $\( L = L_v + L_q \)$. Thus, *even for long video sequences*, the computational cost of SFG is negligible compared to the spiking transformer core.  Moreover, the operations involved in SFG consist of dot products—element-wise in nature—and are computationally lightweight.
>
> **Energy Considerations of SFG**  While cosine similarity involves floating-point arithmetic (element-wise multiplication and summation), it does **not** require floating point matrix multiplications.  Therefore, its energy footprint is **significantly lower** than that of transformer attention layers.  For example, for the 1-bit NF SpikingVTG, considering cost of arithmetic operations on 45nm CMOS, while the transformer core consumes 1.3mJ the SFG mechanism only consumes ~0.01mJ only (Considering a video of 400 secs).
>
> More importantly, the inclusion of SFG leads to **reduced spike activity** across layers (see Figure 3b), translating into **lower computational cost and improved accuracy**. This is highlighted below with an ablation with and without SFG:
>
> | Variant                | @0.5   | @0.7   | mAP   | HIT@1 | Energy (mJ) | Neural Activity |
> |------------------------|--------|--------|-------|--------|--------------|------------------|
> | SpikingVTG w/o SFG     | 64.94  | 47.21  | 40.49 | 67.37  | **15.2**     | 0.41            |
> | **SpikingVTG w/ SFG**  | **67.58** | **50.82** | **40.81** | **68.64** | **13.8**     | **0.34**        |
>
> **Weakness 3**
> > While the authors assert that the model is "well-suited for edge-based deployment," no experiments are conducted on actual edge or neuromorphic hardware. This lack of validation undermines the strength of the deployment claim. All energy efficiency results are estimated via theoretical cost models. The absence of real-device power profiling or runtime measurements limits the practical credibility of the claimed energy savings.
>
> **Response**
>
> In order to validate the feasibility of on-chip deployment, we have implemented our 1-bit (NF)-SpikingVTG to be compatible with Intel Loihi 2 using the Intel LAVA framework. The model is architecturally aligned with Loihi 2’s hardware capabilities, being fully matmul-free (replacing floating-point MACs with integer accumulation operations), removing all non-local normalization layers such as softmax and layer norm. We implement our ternary spiking neurons using **lava.proc.lif.models.PyTernLifModelFixed** and the 1-bit weights are implemented as a custom **ProcessModel** similar to the BitLinear implementation done by [1].
>
> Loihi 2 supports two execution modes: **pipelined** mode for full-input processing and **fall-through** mode for autoregressive token generation. For VTG tasks, we use pipelined mode for offline batch settings (e.g., full video moment retrieval) and fall-through mode for real-time highlight detection (clip-by-clip processing). While [1] implemented a 12-layer matmul-free transformer (370M parameters), our proposed model has only 4 layers with approximately $10\times$ fewer parameters, further reducing memory and compute requirements.
>
> In fall-through mode that is with realtime processing, reported Loihi 2 benchmarks for a matmul-free transformer akin 1-bit NF SpikingVTG, demonstrate a throughput of 41.5 tokens/sec compared to 14.3 tokens/sec for an iso-param conventional transformer on an NVIDIA Jetson GPU (around $3 \times$ speedup). The energy cost per token when we process the entire sequence is 3.7 mJ/token on Loihi 2 versus 11.7 mJ/token on the Jetson GPU, i.e. $>3 \times$ less.
>
> Apart from leveraging computationally efficient accumulation based operations, these improvements stem from Loihi 2’s on-core storage of weights and recurrent states in local SRAM, which minimizes memory movement and reduces both latency and energy consumption.
>
> References:
>
> [1] Abreu, Steven, Sumit Bam Shrestha, Rui-Jie Zhu, and Jason Eshraghian. "Neuromorphic Principles for Efficient Large Language Models on Intel Loihi 2." arXiv preprint arXiv:2503.18002 (2025).
>
> **We thank the reviewer for their constructive feedback, which has helped us strengthen the work. In light of our clarifications, we respectfully request the reviewer to reconsider their rating. We are happy to address any additional questions or comments.**

---

> ### Comment · Reviewer_vi5H · 2025-08-06
>
> Thanks for the authors feedback. The responses address some of my concerns, I also learn some knowledge of SNN in the real applications.
> SNN theoritically has a big potential. The construction of SNN's ego-system does take a while. We should be more encouraging, performance-tolerant and patient for this kind of directions.
> Thus, I would like to lift my rate.

---

> > ### Author Response · Authors · 2025-08-06
> >
> > We sincerely thank the reviewer for the positive feedback and for raising their score. Your thoughtful comments have been truly helpful in strengthening our work.

---

### Author Response · Authors · 2025-08-05

We sincerely thank all the reviewers for their thoughtful comments and constructive feedback. We have carefully addressed each reviewer's comments in our individual rebuttals. As the discussion period is nearing its end, we look forward to engaging further to ensure all feedback and suggestions are thoroughly addressed.

---

### Author Response · Authors · 2025-08-07

We sincerely appreciate the positive comments and constructive feedback from the reviewers. As the discussion period concludes tomorrow, we kindly request any additional comments or concerns you may have—we would be happy to address them.

---

### Note · Authors · 2025-08-11

In our final remarks, we wish to express our sincere gratitude to the reviewers for dedicating their time to evaluating our paper, providing insightful feedback, and thoughtfully considering our rebuttal. We are thankful for their positive reviews and support for our work.

---

### Decision · Program_Chairs · 2025-09-17

**Decision:**

Accept (poster)

**Comment:**

The paper proposes SpikingVTG, a spiking detection transformer for video temporal grounding (VTG) that introduces bio-inspired spiking neural networks for energy-efficient multimodal processing. Key contributions include the Saliency Feedback Gating (SFG) mechanism, Cos-L2 Representation Matching (CLRM) for knowledge transfer, and quantized normalization-free variants suitable for edge deployment. Strengths lie in its novelty, competitive performance on four benchmarks, and significant energy savings, particularly with the 1-bit NF-SpikingVTG. However, the model still underperforms compared to state-of-the-art full-precision VTG models, relies on heavy teacher models, and reports theoretical rather than real-device energy measurements. In rebuttal, the authors provided additional ablations, SFG visualizations, and deployment results on Intel Loihi 2 showing improved efficiency. Despite remaining performance gaps, the paper’s innovation, technical soundness, and potential impact support a borderline accept recommendation agreed by all the reviewers.